# Autologous Transplantation May Still Effectively Treat Relapsed Diffuse Large B-Cell Lymphoma in Selected Patients

**DOI:** 10.3390/cancers15123223

**Published:** 2023-06-16

**Authors:** Reka Rahel Bicsko, Lili Antal, Ferenc Magyari, Róbert Szász, Miklós Udvardy, Arpad Illes, Lajos Gergely

**Affiliations:** Department of Hematology, Institute of Internal Medicine, Faculty of Medicine, University of Debrecen, 4032 Debrecen, Hungaryudvardy.miklosdr@gmail.com (M.U.); lgergely@med.unideb.hu (L.G.)

**Keywords:** diffuse large B-cell lymphoma, relapsed, refractory, autologous stem cell transplantation

## Abstract

**Simple Summary:**

Treating relapsed and refractory diffuse large B-cell lymphoma is still challenging for clinicians. CAR-T cell therapy changed the role of autologous stem cell transplantation in r/r DLBCL patients. The aim of our retrospective study was to assess our data to determine achievable results and to find any additional factors affecting the results of conventional autologous transplantation, determining possible ways to improve the outcome. In a population of 116 DLBCL patients, we found a 75-month median EFS and 105-month median OS. Prognostic markers existing at diagnosis (IPI, LDH, Ann Arbor, COO) lose their significance by the time of transplantation. Consolidative ASCT can be considered an effective and reasonable treatment option for eligible chemosensitive patients with DLBCL, and it still adds survival benefit for additional patients not reaching a complete response before transplantation.

**Abstract:**

Treating relapsed and refractory diffuse large B-cell lymphoma is still challenging for clinicians, but the available CAR-T and bispecific antibodies have revolutionized therapy. Autologous stem cell transplantation was the most effective treatment modality previously. The authors reported data from a single center over ten years. The retrospective study included 116 patients, with 53 relapsed cases, 39 primary refractory cases, 19 who had CNS involvement, and 5 who had received primary consolidation transplants. The median duration of follow-up was 46 months. The median event-free survival was 75 months, and the median overall survival was 105 months for all cases. Five-year overall survival was 59%, and event-free survival was 54%. Pretreatment prognostic factors at diagnosis had no effect on the outcome of transplantation. The authors found no difference between survival in relapsed or refractory cases, and the number of salvage lines or the germinal center/activated B-cell type also did not influence the results. Complete metabolic response before transplantation confirmed by ^18^FDG PET/CT strongly affected survival. The pre-transplant creatinine and CRP levels significantly influenced the long-term outcome. The number of stem cells infused did not affect survival, but engraftment within nine days did result in a longer survival. These data support the finding that the response to salvage therapy did facilitate the identification of a better prognostic group who may still benefit from autologous transplantation.

## 1. Introduction

Diffuse large B-cell lymphoma is one of the most common types of non-Hodgkin’s lymphoma and comprises the largest portion of aggressive lymphomas. Its treatment is still a challenge for clinicians, and a significant proportion of patients still fall into the category of unmet medical need and require salvage therapy with autologous transplantation or chimeric antigen receptor T-cell (CAR-T) therapy. Adding rituximab to CHOP (cyclophosphamide, doxorubicin, vincristine, prednisone) increased the first-line response rate and cure by approximately 10–13% in these patients, but approximately 35–40% of patients relapse or are refractory to this treatment [1]. Adding rituximab to CHOP gave slightly better results in the ABC subgroups, but no other therapy was proven to be beneficial in any subgroup, until polatuzumab–vedotin–R–CHP was investigated in the POLARIX study and achieved a 6.5% better EFS at 2 years in all patients [2]. Salvage treatment of transplant-eligible patients is usually with R-DHAP (dexamethasone, high dose Ara-C, cisplatin) or R-ICE (ifosfamide, etoposide, carboplatin), but recently, it has been shown that CAR-T cells (liso-cel and axi-cel) achieve better results as a second-line treatment compared to conventional salvage chemotherapy [3,4]. Third-line treatment is better with CAR-T cells (tisa-cel, axi-cel, liso-cel) as more patients achieve CR [5,6,7]. The antibody–drug conjugate polatuzumab–vedotin is superior to conventional chemotherapy in second- and third-line salvage settings, making this drug an easily accessible therapeutic choice for most clinicians if it has not already been used as a first-line treatment [8]. The role of autologous transplantation after successful polatuzumab salvage therapy in selected transplant-eligible patients adds a clear survival benefit [9]. However, the question remains as to whether autologous transplantation should follow successful CAR-T salvage therapy. It has been reported that the more lines of therapy are required before transplantation, the worse the long-term results of this treatment [10]. The price and availability of CAR-T therapy make this treatment difficult; thus, a portion of patients still receive conventional chemo–immunotherapy-based salvage therapy in real life, leaving CAR-T therapy as a third- or fourth-line salvage option for non-responding transplant-eligible patients. 

As autologous stem cell transplantation is still the most critical consolidating therapy in r/r DLBCL patients, the authors analyzed their institution’s data to determine achievable results and to find any additional factors affecting the results of this treatment modality, determining possible ways to improve the outcome. CAR-T cell therapy was not available in Hungary at the time interval reported by the authors, so the results reflect what could be achieved by salvage therapies and autologous transplantation.

## 2. Materials and Methods

A retrospective analysis of transplanted diffuse large B-cell lymphoma patients was reported from a single center at the University of Debrecen. All patients were included who underwent autologous transplantation with the diagnosis of diffuse large B-cell lymphoma (DLBCL) between 1 January 2010 and 30 June 2021. The patient’s clinical data were retrospectively collected from the hospital medical records. The data were also cross-checked with the data reported to the European Bone Marrow Transplantation Society (EBMT). Survival times were calculated from the time of transplantation. The censoring event of overall survival (OS) was the patients’ death, and in event-free survival (EFS), either lymphoma progression or death of any cause. Primary refractory cases were defined as not responding after four cycles of first-line chemotherapy determined by interim ^18^FDG positron emission tomography/CT (PET/CT) scan or not achieving CR after first-line chemo–immunotherapy based on end-of-treatment evaluation PET/CT or relapsing within 12 months of receiving first-line rituximab-containing therapy. PET/CT was used to confirm the disease status before and after transplantation.

Salvage therapy was administered to patients until PET was negative. PET positivity was only considered an acceptable result before transplantation, where the salvage treatments were not completely effective, and no other salvage therapy was available at the time; thus, the patient achieved the best possible response with the available therapy. In these cases, tumor reduction had to be at least a partial response. However, more than an 80% tumor reduction was detected in most of the patients not reaching complete metabolic response (CMR) before transplantation. Complete response (CR) refers to CMR in patients’ pre-transplant and post-transplant evaluation, according to the Lugano classification [11]. All patients not fulfilling the criteria for CMR were reported as having PR. Patients with stable disease or progressive disease were not transplanted.

## 3. Patient Characteristics

In the defined time interval, 116 patients with diffuse large B-cell lymphoma (DLBCL) underwent autologous hematopoietic stem cell transplantation (auto-HSCT). All patients who were transplanted during this period were included. There were 11 cases of primary central nervous lymphoma (CNS lymphoma) and 8 cases with systemic plus CNS involvement. These all were transplanted as part of the first-line consolidation therapy. 4 primary mediastinal B-cell lymphoma and 1 very aggressive DLBCL were also transplanted first line. The rest of the patients were relapsed/refractory cases, as 39 primary refractory and 53 relapsed patients were transplanted after responding to salvage treatment. The baseline characteristics of patients at diagnosis are summarized in Table 1. The median age at initial diagnosis was 58 years (range 18–74), and 63 (54.3%) were male. The clinical characteristics at diagnosis were as follows: 93 patients (80.2%) had advanced-stage disease (Ann Arbor stage III–IV); 102 patients (87.9%) had an Eastern Cooperative Oncology Group performance status of 0 or 1; The international prognostic index risk groups were low-risk in 30 (25.8%), intermediate in 39 (33.6%), and high-risk in 47 (40.5%) patients. The cell of origin based on the Hans algorithm was used to define the histological groups of patients, but it was only routinely used after 2015. Germinal center lymphoma was diagnosed in 19 cases (16.3%), non-germinal center B-cell-like lymphoma was diagnosed in 36 cases (31%), and 61 (52.5%) patients were classified as “not otherwise specified”. Most patients received R-CHOP as their first-line treatment. CNS lymphoma cases were treated with high-dose methotrexate–rituximab–vincristine (R-MPV) and were transplanted as part of the first-line treatment. Relapsed/refractory disease was confirmed by ^18^FDG-PET/CT scan. Re-biopsy was not routinely performed, and this was only carried out in cases where late relapse or unusual localization of the relapse was detected. The most common salvage regimen was R-DHAP (rituximab, cisplatin, dexamethasone, and high-dose cytarabine, n = 72), followed by R-ICE (rituximab, ifosfamide, carboplatin, and etoposide, n = 16). Patients received an average of 2.22 cycles of salvage chemotherapy before transplantation. The response was evaluated after two cycles of salvage therapy, and patients who did not respond were switched to a different salvage therapy. Forty-two patients received more than one type of salvage chemotherapy. Radiotherapy was used in two cases before transplantation and in four cases early after transplantation as part of consolidation therapy. Most patients (59.5%) achieved complete metabolic remission before transplantation, but the others failed to do so. In their case, auto-HSCT was performed for those in partial remission. All primary consolidations were carried out in CR (n = 22). The conditioning regimen was R-BEAM (rituximab, carmustine, etoposide, cytarabine, and melphalan) for DLBCL (n = 92) and TBC (thiotepa, busulfan, and cyclophosphamide) for CNS lymphoma, lymphoma with CNS involvement, and selected aggressive cases where the CNS propagation risk was high (n = 24). According to the guidelines, the target number of infused viable CD34+ stem cells was 5 × 10^6^/kg body weight at the institution. 

### Statistical Analysis

The Kaplan–Meier method analyzed and compared unadjusted survival distributions using the log-rank (Mantel–Cox) test. Hazard ratios were calculated using the log-rank test. The effect of variables on the outcome was investigated using the ROC analysis to define cut-off points where needed. Cut-off values were always rounded to the nearest non-decimal number, except where noted. Two-sided *p*-values of <0.05 (5%) were considered to be statistically significant. Statistical analyses were performed using GraphPad Prism v9.5 (GraphPad Software, San Diego, CA, USA) and SPSS v 28.0 software (IBM Corp. Armonk, NY, USA). Survival graphs were created using GraphPad Prism v9.5 software.

## 4. Results

One hundred and sixteen patients with DLBCL were transplanted at the center. There were 53 relapsed cases where the average time between initial diagnosis to transplantation was 42 months. 39 cases were primary refractory, and 24 patients received a first-line consolidation transplant, primarily due to central nervous system involvement. The median age at the time of transplantation was 58 years (range 18–74). The median duration of follow-up was 46 months (8 to 133). The median OS was 105 months for the total population, while the median EFS was 75 months. (Figure 1).

### 4.1. The Effect of Initial Prognostic Factors Established at Diagnosis on the Outcome of Transplantation

The authors analyzed the transplantation outcome data based on prognostic factors that were calculated at the time of DLBCL diagnosis. No significant difference was found in EFS between the low- and intermediate-/high-risk groups based on the IPI score (data not presented). The median event-free survival was 93 months in the IPI 0–2 group and 43 months in the IPI 3–5 group (*p* = 0.19). Elevated LDH values at diagnosis did not affect the survival results (data not presented). Median EFS was 93 months in the absence of B symptoms, while it was 25 months in the presence of B symptoms (*p* = 0.12); however, due to data variation, this was not statistically significant. According to the Hans algorithm, no significant differences were found in survival between the histological subgroups of CG versus non-CG. The authors compared the groups with primary and secondary CNS involvement because CNS involvement is associated with a worse prognosis. Median event-free survival was 26.5 months in the systemic plus CNS lymphoma group and not reached (*p* = 0.12) in the primary CNS group. EFS curves reached a stable plateau in the latter cluster at 80.8% (Figure 2). A detailed summary of hazard ratios associated with different factors is listed in Table 2.

### 4.2. Effect of Pre-Transplantation Prognostic Factors on Outcome

The authors found no difference in event-free and overall survival between primary refractory and relapsed patients (*p* = 0.96), and no disparity according to whether the transplant was performed as primary consolidation, or after relapse or primary refractoriness (*p* = 0.7, Figure 2); the median EFS was not reached vs. 105 vs. 75 months, respectively. Survival was not affected by the number of lines of prior salvage chemotherapy either; median event-free survival was 93 months in the case of one type of salvage therapy and 21 months when patients received more than one line of salvage therapy (*p* = 0.13). Patients who achieved a PET-negative complete response before transplantation had a significantly better 5-year overall survival than those who achieved only partial remission (62.5% vs. 30%, *p* = 0.0009). Median overall survival was 36 months for the PET-positive group and not reached for the PET-negative group (Figure 3). Bone marrow function before transplantation, such as white blood cell count (WBC) or hemoglobin level, had no significant effect on survival results. Median EFS was 75 months in the normal WBC group and not reached in the case of a low white blood cell count (*p* = 0.5). Neither the absolute lymphocyte and monocyte count nor the lymphocyte-to-monocyte ratio significantly affected survival at this time point. Median EFS was 69 months if the absolute lymphocyte count was above 1.1 G/L and not reached in a group with a lower absolute lymphocyte value (*p* = 0.6). Interestingly, renal function did affect the outcome of transplantation, as both higher (>90 umol/L) creatinine and higher (>4.5 mmol/L) blood urea nitrogen levels were associated with a significantly worse survival. Median EFS was 93 vs. 13 months (*p* = 0.028) in groups with normal and elevated creatinine levels (Figure 4). The difference was also reflected in overall survival as patients with high urea levels had a median EFS of 46 months in the patients with higher blood urea nitrogen levels and was not reached for the lower urea group (*p* = 0.018). In comparison the median OS was 74 months vs. not reached in these groups (*p* = 0.012). The presence of ongoing infection did defer patients from transplant. However, patients with slightly elevated CRP levels without obvious signs of infection were allowed to proceed to transplantation. Elevated C-reactive protein levels measured directly before transplantation also significantly affected the results. A CRP level over 6 mg/L was associated with a significantly shorter EFS and OS. Median EFS was 105 months in the group with normal CRP level and 35 months with an elevated CRP level (*p* = 0.04), while median OS was not reached vs. 36 months (*p* = 0.038), respectively. As per local guidelines, the target stem cell number infused was 5 million CD34 per body kg of patients. The actual number of stem cells administered varied between 2.68 and 9.69 × 10^6^ per kg body weight (median: 5.84). The number of stem cells infused during transplantation did not affect survival. Engraftment occurred at a median of 9.4 days (7–18) after stem cell infusion. However, faster engraftment resulted in a better outcome, as median event-free survival was not reached vs. 36 months (*p* = 0.025), and median overall survival was not reached vs. 41 months (*p* = 0.01) in cases of engraftment within 9 days, compared with over 9 days (Figure 5). See Table 2 for hazard ratios associated with the different factors investigated.

## 5. Discussion

Autologous peripheral stem cell transplantation is still an effective therapeutic modality among patients with relapsed or refractory diffuse large B-cell lymphoma showing chemosensitivity to salvage therapy [12]. In this single-center analysis, the authors presented survival outcomes, exploring variables associated with survival during this therapeutic modality.

The long-term outcome of the patients reported is similar to those in previously published studies, resulting in more than a 50% long-term survival [13]. The conditioning consisted of rituximab and BEAM conditioning in the cases reported. Thus, the reports reflect the R-BEAM conditioning results. The authors had a good experience with this protocol, with acceptable toxicities. As several prognostic factors at diagnosis have a clear role in the initial prognosis of DLBCL patients, the authors examined whether these factors still play a role for transplanted patients. The authors found no difference in outcomes after autologous stem cell transplantation for primary refractory and relapsed DLBCL based on the cell of origin. This is somewhat contrary to the previously published results, as the germinal center type was reported to have a worse long-term survival [14]. However, a similar outcome has been reported in another publication, where no difference was found according to the cell of origin [15,16].

The IPI at diagnosis did not differentiate prognostic groups in our cohort, which is also contrary to previously published results, as higher IPI patients had an inferior survival after transplant [17]. It has been reported that IPI before transplantation may impact outcomes, but it was not evaluated in the cases reported in this paper [18]. LDH level and B symptoms at the time of diagnosis did not significantly affect the transplantation outcome either. This data corresponds with the previously published data. The reported results in this study support the fact that the clinical behavior of the disease, especially the response to salvage therapy being a strong biological prognostic factor, overwrites all pretreatment prognostic factors. Survival data were not affected by whether the transplant was due to relapse or primary chemo-refractoriness. Only the response to salvage therapy was important, which also determined whether the patient could proceed to transplant. The results presented here are somewhat different from the published data, as relapsed patients are reported to have a better prognosis than refractory cases [19]. The authors found no difference concerning whether a patient responded to the first choice of salvage therapy or required additional lines of salvage therapy. No matter how many cycles of therapy were required, if the patient achieved a good response, the transplant was beneficial. This is a significant finding. However, the quality of remission seen on PET/CT before transplantation greatly influenced the results. This has already been published, and the authors’ data also support this important finding [20,21,22]. However, autologous transplantation can still benefit patients responding to salvage therapy but not reaching a complete metabolic response [23]. The question regarding whether these patients require additional novel therapy to reach remission or whether the novel treatments can be reserved for cases with relapse after the transplantation still remains unanswered [24]. Bone marrow function before transplantation did not prove to be of decisive importance, as neither the hemoglobin level at the time of transplantation nor the white blood cell count and lymphocyte-to-monocyte ratio affected survival data. The later was shown to have a clear prognostic role at the beginning of therapy [25]. There was a tendency for better outcomes with platelet levels above 100 G/L, but this was only significant for overall survival. The authors found no correlation between the number of stem cells infused and survival. However, the time required for engraftment significantly influenced long-term outcomes. This unique data must be further explored, as previous treatments did not influence this. The higher-than-normal CRP value without ongoing infection, as well as the elevated blood urea nitrogen and creatinine levels, significantly influenced the results. This highlighted that, besides response to therapy being a strong biological prognostic factor, these additional factors characterize the biological background, having prognostic influence in these patients.

The long-term results with CAR-T salvage therapy are good. The ORR was 53% and 39% reached CR from the treated patient group, and the ORR was 39.5% in the intention-to-treat population in a previous study [26]. These data are similar to the results presented in this paper, but the patients included only transplanted chemosensitive cases, whereas CAR-T achieved a response in more patients, resulting in a better EFS and OS in the intention-to-treat population. The major obstacle of CAR-T therapy is the waiting time for the production of the T-cells. The results of axi-cel therapy were published in the ZUMA-1 trial and comparing these results with the older SCHOLAR-1 trial demonstrated an overall survival benefit of an additional 20% at 2 years in the CAR-T arm [27]. These data strongly favor CAR-T therapy over autologous bone marrow transplantation, as was also investigated in a systematic meta-analysis showing the superiority of CAR-T [28]. However, the data presented in this paper outlines that a certain population of patients can still benefit from autologous transplantation, especially if CAR-T is not universally available.

Several limitations must be considered in the interpretation of these results. First, the retrospective design is subject to the inherent selection bias of non-randomized retrospective data. Furthermore, there was missing information for some patients who were lost to follow-up, a long study period with different eras of diagnostic criteria, as well as therapy with potentially different management approaches. Our data set had some missing data, which affected mainly the DLBCL subtypes according to the Hans algorithm (germinal center B-cell-like lymphoma, non-germinal center B-cell-like lymphoma, not otherwise specified). The results of this study were also limited by the relatively small cohort of heterogeneous patients, the heterogeneity of prior chemotherapy, and distinct causes of transplantation, such as first-line consolidation, and refractory and relapsed cases, preventing us from comparing protocol efficacy and drawing definitive conclusions. However, this heterogeneity also provided some basis for comparison between different patient groups.

## 6. Conclusions

In conclusion, consolidative ASCT can be considered an effective and reasonable treatment option for eligible chemosensitive patients in DLBCL when CAR-T is not universally available, or the patient responds to salvage therapy [29]. With careful patient selection, the results are acceptable. With the availability of novel treatment options such as CAR-T cells and bispecific antibodies, further studies are required to better understand how to sequence these treatment modalities and determine the role of autologous transplantation [30]. The authors’ findings support other studies reporting that complete metabolic remission before autologous transplantation is associated with comparable overall survival to CAR-T therapy. The data also supports the important fact that patients without documented CR can still benefit from transplantation. Prognostic markers existing at the diagnosis lose their significance by the time of transplantation.

## Figures and Tables

**Figure 1 cancers-15-03223-f001:**
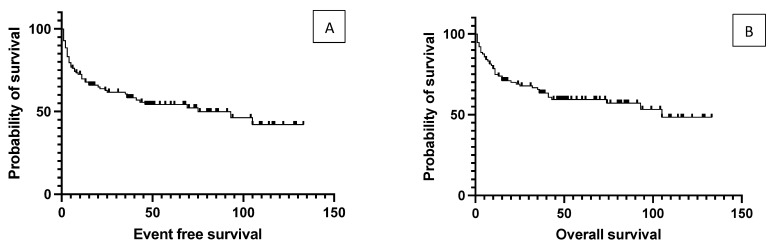
Event-free survival (**A**) and overall survival (**B**) of the total study population. Median event-free survival was 75 months, and median overall survival was 105 months.

**Figure 2 cancers-15-03223-f002:**
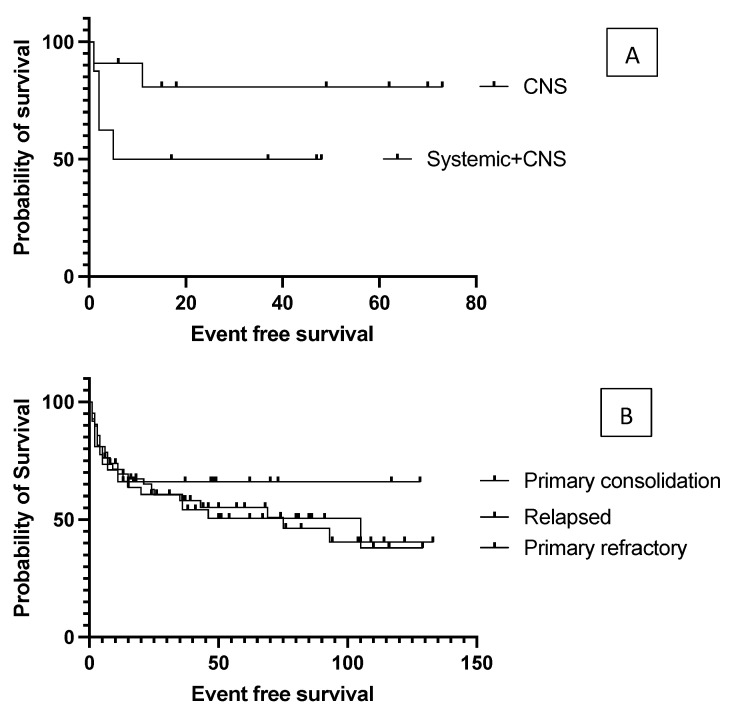
(**A**) Event-free survival in primary CNS lymphoma versus systemic plus CNS involvement. Median event-free survival was 26.5 months in systemic plus CNS lymphoma group, and not reached in the primary CNS group, but the difference was not statistically significant. CNS = central nervous system. (**B**) Event-free survival data according to whether the transplant was performed as primary consolidation or after relapse or primary refractoriness. No significant difference was found between these groups.

**Figure 3 cancers-15-03223-f003:**
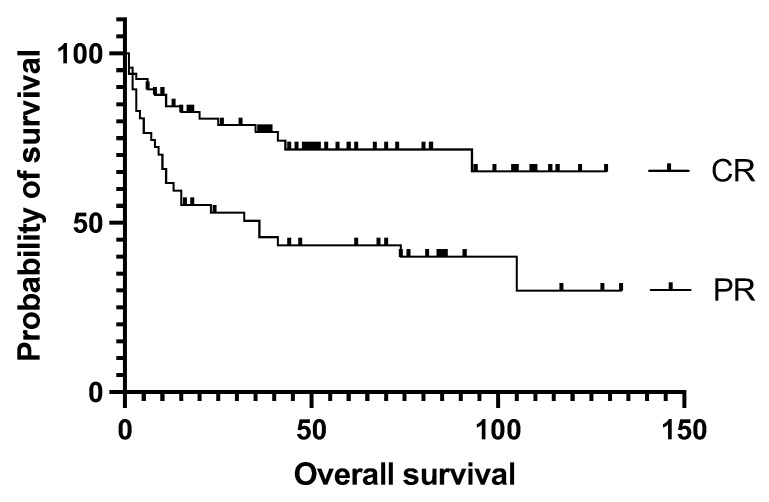
Overall survival data based on the quality of remission seen on PET/CT before transplantation. Median OS was 36 months in PR group, and not reached in CR group. Survival curves reached a stable plateau at 65.2% in CR. CR = complete metabolic remission, OS = overall survival, PR = partial remission.

**Figure 4 cancers-15-03223-f004:**
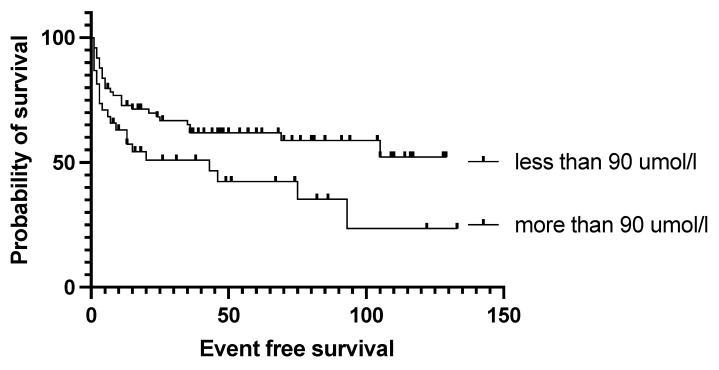
Event-free survival data based on creatinine level with a 90 umol/L cut off. Higher creatinine level was associated with significantly shorter survival.

**Figure 5 cancers-15-03223-f005:**
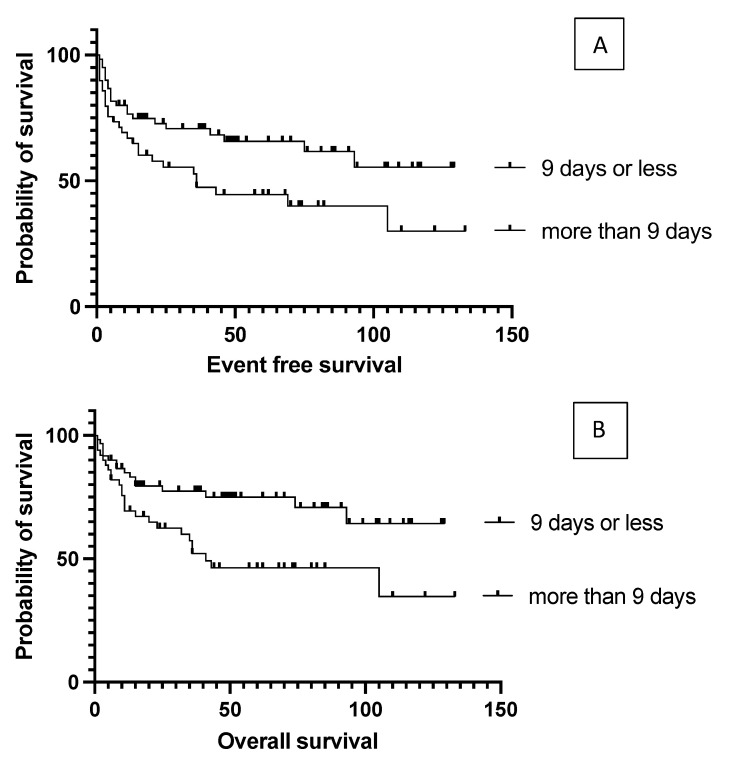
Event-free (**A**) and overall (**B**) survival data according to the time required for engraftment. Faster (within 9 days) engraftment resulted in significantly better outcome.

**Table 1 cancers-15-03223-t001:** Patient characteristics.

Characteristics	Variable	Value
Sex	Male	63 (54.3%)
	Female	53 (45.7%)
Age at diagnosis (years)	Median	55
	Range	17–73
Subtype DLBCL	Germinal center	19 (16.3%)
	Non-germinal center	36 (31%)
	Not otherwise specified	61 (52.5%)
PCNSL		11 (9%)
B symptoms	Present	43 (37%)
	Absent	73 (63%)
Ann Arbor stage at diagnosis	I–II	23 (19.8%)
	III–IV	93 (80.2%)
IPI at diagnosis	0–1	30 (25.8%)
	2	39 (33.6%)
	3–5	47 (40.5%)
ECOG PS	0–1	102 (88%)
	2	14 (12%)
Age at auto-SCT (years)	Median	58
	Range	18–74
Lines of therapy before auto-SCT	1	16 (13.8%)
	2	58 (50%)
	3	40 (34.5%)
	4	2 (1.7%)
Response before auto-SCT	Partial response	47 (40.5%)
	Complete response	69 (59.5%)
Conditioning regimen	R-BEAM	92 (79.4%)
	TBC	24 (20.6%)
Number of stem cells administered	Range (×10^6^/bwkg)	2.68–9.69
Time to engraftment (days)	Range	7–18
	Average	9.4

DLBCL—diffuse large B-cell lymphoma, ECOG PS—Eastern Cooperative Oncology Group, performance status, IPI—international prognostic index, PCNSL—primary central nervous system lymphoma, SCT—stem cell transplantation.

**Table 2 cancers-15-03223-t002:** Factors affecting the outcome of autologous transplantation in diffuse large B-cell lymphoma patients. Overall survival (OS) and event-free survival (EFS) were calculated from the date of transplant.

Univariate Analysis				
	EFS		OS	
	Hazard Ratio	*p*-Value	Hazard Ratio	*p*-Value
Factors at diagnosis				
Ann Arbor stage III–IV	2.254	0.1053	1.728	0.2435
IPI 3–5	1.425	0.1934	1.411	0.2412
Bulky disease	1.385	0.4724	1.042	0.9140
B symptoms	1.922	0.0760	1.606	0.1006
CG histology	0.765	0.5259	0.887	0.7456
Normal LDH	0.810	0.5468	0.828	0.6033
Factors at transplantation				
PET-negative	0.422	0.0015	0.382	0.0010
More than one salvage therapy	1.527	0.133	1.599	0.1158
Primary refractory disease	1.014	0.9624	1.024	0.9566
Engraftment before 9 days	0.531	0.0253	0.452	0.0101
More than 4 × 10^6^/kg CD34+ graft	0.684	0.2195	1.031	0.9402
Hgb over 100 g/L	0.89	0.7276	0.779	0.4666
Abs. lymphocyte over 1 G/L	1.078	0.8176	1.212	0.5321
Thrombocyte over 100 G/L	0.704	0.2103	0.491	0.0241
CRP less than 6 mg/L	0.558	0.0453	0.530	0.0382
Creatinine below 90 umol/L	0.504	0.0288	0.550	0.0408
BUN less than 4.5 mmol/L	0.437	0.0184	0.375	0.0121

Significant difference was stated where the *p*-value was less than 0.05. BUN—blood urea nitrogen, CRP—C reactive protein, PET—positron emission tomography, Hgb—hemoglobin, CG—germinal center type by Hans algorithm, IPI—international prognostic index, LDH—lactate dehydrogenase enzyme.

## Data Availability

Third-party data restrictions apply to the availability of these data. Data were obtained from patients and are available from the authors with the permission of the patients, as no consent was obtained beforehand for data sharing.

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
