# Peer review of "Autologous Transplantation May Still Effectively Treat Relapsed Diffuse Large B-Cell Lymphoma in Selected Patients"

_cancers, 2023, doi:10.3390/cancers15123223_

Round 1
Reviewer 1 Report
In this paper, the Authors report on the clinical outcome of DLBCL patients who received ASCT at their institution in a timespan of ten years. Their results confirm that achieving a complete response before transplantation is the strongest prognostic factor. Prognostic factors that are considered relevant at diagnosis such as IPI or COG did not show any impact in survival in this cohort.
I think that the most relevant information in this cohort is that a not negligible proportion of patient failing to achieve CR before ASCT still benefits from the procedure.
Although the role of ASCT as consolidation treatment’s in chemosensitive is a matter of debate in the CAR-T era, I fear that overall this paper does not add significant novel information.
Weaknesses
Heterogeneous cohort, some patient received ASCT as first line treatment and some of them had PMBCL or PCNSL
Retrospective analyses
Most of the reported results have already been reported
The English should be revised.
Author Response
Answer for reviewer 1 questions:
Thank you for reviewing this paper and highlighting the weaknesses.
Thank you for emphasizing what we also found that a significant portion of our patients were not able to achieve CR before transplant (it was in the era of conventional salvage chemotherapies) and proceeded to transplant with still acceptable results.
Indeed, this was a heterogenous population, but it provided means for comparison between different patient groups as we presented.
We strongly agree that the role of autologous transplantation in the CAR-T era is a debate. That is one of the reasons we provide our data. We did not have CAR-T cell therapy in the period reported, so our data is forming a non-CAR-T treated population.
We have expanded the introduction and added additional CAR-T data comparison and references, to reflect the debate and showing the superiority of CAR-T salvage over conventional salvage therapy.
The English has been checked and corrected.
Kind regards,
Reka Rahel Bicsko
Reviewer 2 Report
It would be worth noting the sequence of CAR_T vs Auto with first line approval of Pola. It is a good study.
Author Response
Answer for reviewer 2:
Thank you for reviewing this paper.
As per your recommendation (also other reviewers recommended it), we have expanded the CAR-T section in discussion.
We have added the POLARIX first-line data into introduction, and added that the recommended 2nd line salvage is clearly CAR-T in r/r DLBCL. Also noted that this trial was done in Hungary when there were no CAR-T cells available, making a unique opportunity to use novel salvage treatment without CAR-T and examine the results of conventional transplantation.
Kind regards,
Reka Rahel Bicsko
Reviewer 3 Report
This is an interesting retrospective, single-centre study assessing the outcomes of autologous stem cell transplant (ASCT) in patients with relapsed/ refractory diffuse large B-cell lymphoma (DLBCL). The data are clearly presented in the manuscript, tables and figures. It may add information to the literature on the outcome ASCT in patients with refractory/relapsed DLBCL.
There are some suggestions for the authors.
1. The authors may mention or discuss whether CAR-T therapy is available at their center for the treatment of refractory DLBCL.
2. The authors may compare their survival results to the published long-term outcomes of CAR-T therapy for lymphoma or DLBCL.
Author Response
Answer for reviewer 3 questions:
Thank you for reviewing this paper.
We have added the additional information into the manuscript according to reviewers’ suggestions.
We have included a sentence into the introduction, that CAR-T therapy was not available in Hungary during the period investigated in this trial. This makes it a unique data set, as we could use other salvage options (polatuzumab etc.) but not the CAR-T.
In the discussion section the CAR-T data is presented in more detailed way and compared to our dataset as per your recommendation. Also noted the problem with CAR-T data, that it should be interpreted on the ITT population. Made a clear comment that CAR-T salvage is superior to conventional salvage chemotherapy and is the recommended 2nd line treatment.
Kind regards,
Reka Rahel Bicsko
Round 2
Reviewer 1 Report
The Authors have addressed all my concerns.
The English Language has significantly improved from previous version